# A Probe-Based qPCR Method, Targeting 16S rRNA Gene, for the Quantification of *Paenibacillus larvae* Spores in Powdered Sugar Samples

Elena Carra [1,*], Giorgio Galletti [1], Emanuele Carpana [2], Federica Bergamini [1], Giulio Loglio [3], Filippo Bosi [4], Stefano Palminteri [5] and Stefano Bassi [1]

1   Istituto Zooprofilattico Sperimentale della Lombardia e dell'Emilia Romagna "B. Ubertini", 25124 Brescia, Italy
2   CREA Research Centre for Agriculture and Environment, 40128 Bologna, Italy
3   Dipartimento di Prevenzione Veterinario, Agenzia di Tutela della Salute di Bergamo, 24121 Bergamo, Italy
4   Dipartimento di Sanità Pubblica, Azienda Unità Sanitaria Locale della Romagna, 48124 Ravenna, Italy
5   Dipartimento di Sanità Pubblica, Azienda USL di Bologna, 40124 Bologna, Italy
*   Correspondence: elena.carra@izsler.it

**Abstract:** *Paenibacillus larvae* (*P. larvae*) is responsible for American foulbrood (AFB), the most severe bacterial disease of honeybees. The enumeration of *P. larvae* spores in substrates taken from hives allows for the identification of the contamination levels of the colonies, mostly in those with atypical symptoms or with asymptomatic infections; in these cases, it is essential for the effective control of American foulbrood (AFB). In this work we described a new quantitative TaqMan® probe-based real-time PCR (qPCR) assay, targeting the 16S rRNA gene of *P. larvae*, used for the quantification of *P. larvae* spores in powdered sugar samples collected from hives, in comparison to the culture. A total of 105 colonies were selected, belonging to 10 apiaries with different levels of infection, located in northern Italy. The proportions of positive colonies was 54% (57/105) with the culture and 66% (69/105) with qPCR. A significant difference between the two methods was found with McNemar's test (p = 0.02). Out of the 51 positive samples by both methods, 45 showed higher infection by qPCR compared to the culture. A close concordance with the clinical–epidemiological status of the hives was observed by both methods, with higher infection levels found by qPCR.

**Keywords:** American foulbrood; quantitative TaqMan® real-time PCR (qPCR); culture method

## 1. Introduction

*P. larvae is* a Gram-positive bacterium forming endospores. It is responsible for American foulbrood (AFB), the most widespread and destructive bacterial disease that affects only the larval and pupal stages of honeybee broods of *Apis mellifera* and other *Apis* spp. The bacterium produces over one billion spores in each infected larva. The spores are long-lived and extremely resilient to heat and chemical agents. They can survive for many years in scales (from dead brood), hive products, and equipment. Only the spores are capable of inducing the infection [1,2].

The disease occurs throughout the world, causing considerable economic losses to beekeepers [3]; clinically, the transformation of the dead larvae into a ropy mass is the peculiar feature, which forms a characteristic viscous thread if a matchstick is inserted into the cell and then pulled out. The disease often causes colony death if left untreated once clinical symptoms have emerged [4].

AFB is a notifiable disease, with few treatment options [5].

The detection of AFB is based on the recognition of its typical clinical symptoms. Pathogens can be identified by means of laboratory procedures [2]. Techniques for the identification of the microorganism are based on microbiological characterization, biochemical profiling, antibody-based techniques, microscopy, and the polymerase chain reaction (PCR) [2].

Diagnostic tools are used allowing to assess early and quantify the *P. larvae* spores in a broad range of hive samples such as honey and hive debris. Culture and quantitative PCR are the principal methods used for the quantification of *P. larvae* spores in hive materials. Culture-based methods suffer from sample pre-treatment procedures, the choice of culture media affecting the germination of the spores, and from the *P. larvae* genotype. Culture-based methods are time-consuming and require species confirmation by biochemical tests, microscopic techniques and/or MALDI-TOF mass spectrometry [2].

*P. larvae* strains have been genotyped by several techniques, including repetitive element PCR (REP-PCR), pulsed-field gel electrophoresis (PFGE), multilocus sequence typing (MLST), and multilocus variable number of tandem repeat analysis (MLVA) [1,6,7].

Repetitive element PCR (rep-PCR), based on the use of enterobacterial repetitive intergenic consensus (ERIC) primers, revealed five different ERIC genotypes, designated *P. larvae* ERIC I–V, showing that the resulting classification correlates with phenotypic differences, above all in colony morphology and in virulence [1,8]. The genotypes ERIC I and ERIC II have a practical importance and are regularly isolated from AFB outbreaks in Europe as well as worldwide [1,9–13], while the ERIC III, ERIC IV, and ERIC V genotypes are isolated only very rarely in the field, and they exist as a few isolates in culture collections [5,8].

The genotype ERIC II colonies are distinguished from those of the genotype ERIC I by a particular morphotype and orange color [1,12,14]. Orange-colored Bacillus larvae strains were described in the past by Križanová et al. (1988) [15] and Drobníková et al. (1994) [16] as causes of AFB outbreaks. The authors described atypical symptoms as the following: diseased brood cells with not darkly colored caps; light brown or grey decaying larvae with a watery consistency instead of a glue-like one; and light brown or grey scales that are easily removed from the cells.

*P. larvae* ERIC II strains are rather faster killers than ERIC I ones, being more virulent on the larval level. All infected larvae die within few days, and the majority before cell capping, such that adult bees easily remove the dead larvae from the hive [17]. At the colony level, it can be very difficult to spot the decaying larvae, and the number of perforated, concave, and darkened cappings is much lower than that observed in infections caused by ERIC I genotype [18,19].

In all of *P. larvae* ERIC II AFB cases clinical diagnosis is more laborious, with the risk of missing them, especially in the colonies with asymptomatic infections. AFB generally occurs in clinical form when a certain level of *P. larvae* spore contamination is reached in honeybee colonies [20]. A relatively large number of spores can be present over several seasons in some colonies without evidence of clinical signs of the disease [21–24]. Subclinical infections can lead to relapses of clinical forms in apiaries and promote the horizontal spread of AFB from one hive to another [25].

The presence of *P. larvae* spores and their quantification in hive materials allows for the identification of the contamination levels of the colonies, mostly in those with atypical symptoms or with asymptomatic infections; in these cases, it is decisive for reducing the impact of AFB.

The search for the *P. larvae* spores load in adult bees and hive materials such as honey, wax, and wax debris is described in standard methods, based on microbiological and biomolecular techniques [2,26,27].

Recently, a powdered sugar examination was described by Bassi et al. (2022) [28] as an easy-to-use, nondestructive, and practical material for the assessment of *P. larvae* infection levels in honeybee colonies.

Many molecular methods based on conventional PCR [29–36] or on real-time PCR [37–44] have been previously described for the presence and/or quantification of *P. larvae* infection levels in different hive materials. Quantitative PCR assay (qPCR) is a time- and cost-effective choice with respective to the quantification of *P. larvae* spores and overcomes the limitations of the traditional methods.

Among qPCR methods described in the literature, to our knowledge, only three have been used for the assessment of *P. larvae* spores in hive debris and/or honey samples.

Two assays are designed to target the 16S rRNA gene and use SyBr technology, which suffers from important disadvantages [39,42]. The 16S rRNA gene is a useful target for molecular diagnostic purposes for two main reasons: (a) for being highly conserved among bacterial pathogens, including *P. larvae* strains, although single-nucleotide polymorphisms are known in the sequences of *P. larvae* of different geographic origin and/or genotype; and (b) for being present in eight copies per genome of *P. larvae*, allowing for increased sensitivity of diagnostic assays. These features are considered for the development of diagnostic assays for many bacterial pathogens, including the *P. larvae*.

The third assay is a new TaqMan® probe-based real-time qPCR protocol whose target is the single-copy chromosomal metalloproteinase gene. It was developed for the detection and quantification of spores in hive debris and honey samples [44].

The TaqMan® probe-based real-time qPCR protocol overcomes the limitation of the qPCR based on the use of SyBr technology, not requiring post-PCR processing step, such as the analysis of the melting curves [45]. TaqMan® probes, also known as "fluorogenic 5′ nuclease chemistry", are widely used in many diagnostic real-time PCR assays. Specificity is much higher with the use of specific probes besides primers [45]. The availability of these fluorogenic probes enabled the development of a real-time method for detecting only specific amplification products. Specific hybridization between probe and target is required to generate fluorescent signal. The TaqMan® probe can be labeled with many different fluorophores useful in multiplex assays: probes can be labeled with distinguishable reporter dyes, which allows amplification and detection of two or more distinct sequences in one reaction tube. Post-PCR processing is eliminated, which reduces assay labor and material costs.

The aim of the present work was to describe a new quantitative TaqMan® probe-based real-time PCR (qPCR) assay, targeting the 16S rRNA gene of *P. larvae*, in comparison to the culture. The two methods were employed for the assessment of *P. larvae* spores in samples of powdered sugar taken from hives, to evaluate the *P. larvae* infection levels in honeybee colonies with different levels of infection. The results of the comparison between these methods are described and discussed.

## 2. Materials and Methods

### 2.1. Clinical Inspection and Sample Collection

A total of 105 colonies were selected for the present work. They were collected from 10 apiaries located in two regions of northern Italy: Lombardy and Emilia-Romagna. The apiaries had different histories of AFB's presence. In six apiaries (*Group A*: 60 colonies), recurrent outbreaks of AFB were detected in the years prior to this work, while in the other four apiaries (*Group B*: 45 colonies) there were no cases of AFB in the last 2 years, and for this reason they were selected as the control group. Each colony was checked for clinical signs as described in Bassi et al., 2022 [28], before sampling.

The samples of powdered sugar were collected as described in Bassi et al. (2022) [28].

In *Group A*, 15 samples were collected from symptomatic colonies (*subgroup A-1*) and 45 from colonies without symptoms of the disease (*subgroup A-2*). In *Group B*, 45 samples of powdered sugar were collected.

### 2.2. Microbiological Analysis

The method was done exactly according to Bassi et al., 2022 [28], starting from 1 g of collected sugar. In brief, for each sample, 1 g of sugar was dissolved by being shaken in a 15 mL test tube containing 9 mL of sterile distilled water. Then, it was heated at 85 °C for 15 min, obtaining a "pre-treatment suspension", which was plated on Mueller–Hinton broth, Yeast extract, Potassium Phosphate, Glucose, and Pyruvate (MYPGP) agar, incubated at 37 °C in an atmosphere with 10% $CO_2$. The plates were examined after 3 and 8 days [28].

Colonies of *P. larvae* were identified based on growth time and morphological characteristics. In the positive samples, the colonies always resulted uniform in the morphology. One to five colonies per sample, with a *P.-larvae*-like morphology, were subjected to a Gram stain and catalase reaction, always resulting in Gram-positive rods and catalase negativity.

One strain per sample was sub-cultured on a Tryptone Soy Yeast Extract Agar (TSYEA) slant and confirmed by PCR as previously described [30].

When the number of *P. larvae* colonies was very large, to prevent the colony counting, tenfold dilutions were prepared from the pre-treatment suspension and then cultured as described above.

The number of viable spores was calculated and expressed as colony-forming units (CFUs). The limit of detection (LOD) of the method was 20 CFU/g.

### 2.3. Molecular Analysis

### 2.3.1. DNA Extraction

Two mL of each sample pre-treatment suspension (see the above paragraph), corresponding to 0.2 g of a sugar sample, was centrifuged at $21,000 \times g$ for 10 min. The obtained pellet was washed with 1 mL of a solution containing 10 mM Tris-HCl, 1 mM EDTA, and NaCl 0.9%, centrifuged at $21,000 \times g$ for 10 min, resuspended in 200 μL of Lysozyme solution, and incubated for 30 min at 37 °C according to the protocol previously described [32]. Then, 10 μL of Internal Control (IC) High Concentration (Qiagen, Hilden, Germany) and 20 μL of proteinase K (20 mg/mL) were added to each sample and incubated at 56 °C for 1–3 h until the complete resuspension of the pellet. Then, DNA purification was performed according to Exgene $^{TM}$ Tissue SV kit (GeneAll® Biotechnology Co., Seoul, Korea). At the end, DNA was eluted in 100 μL of AE elution buffer (10 mM tris-HCl pH 9.0; 0.5 mM EDTA) according to the manufacturer's instructions in the kit. Extracted DNA was quality checked spectrophotometrically using the Synergy$^{TM}$ HTX (BioTek Instruments, Inc., Winooski, VT, USA) by measuring the absorbance at wavelengths 260/280 nm.

### 2.3.2. TaqMan® Real-Time PCR Assay for the Detection of *P. larvae* DNA in Sugar Samples

To perform a highly specific and sensitive PCR, a new set of primers and a Taqman® probe (Table S1) were designed and used in the present study. They were designed using the Primer Express® software v.3.0 (Applied Biosystems, Foster City, CA, USA) according to the 16S rRNA gene sequences of *P. larvae* strains of different ERIC genotypes available from GenBank (Table S2) in order to assess in silico inclusivity. The sequences were aligned by Clustal W, as implemented in BioEdit v.7.0.8.0 [46]. The specificity of the primers and probe was assessed in silico (exclusivity) via a BLAST search against 16S rRNA gene sequences of potential hive bacterial contaminants belonging to *Paenibacillus*, *Bacillus*, and *Enterococcus* genera available from GenBank (Table S3).

The specificity of the primers and probe were additionally tested by real-time PCR against DNA extracted from 17 different bacteria species: *Paenibacillus alvei*, *P. thiaminolyticus*, *P. amylolyticus*, *P. azotofixans*, and *P. apiarius*; *Bacillus thuringiensis*, *B. cereus*, *B. sphaericus*, *B. licheniformis*, *B. subtilis*, *B. pumilus*, *B. megaterium*, *B. circulans*, *B. firmus*, and *B. polimixa*; *Brevibacillus laterosporu* as well as *Enterococcus faecalis*.

All of the *P. larvae* detection real-time PCR (real-time PCR) assays were performed with a QuantiFast Pathogen PCR + IC kit (Qiagen, Hilden, Germany) according to the manufacturer's instructions. The amplification reactions were carried out in a final volume of 25 μL containing 5 μL of DNA extract, 1 X QuantiFast Pathogen PCR Master Mix with ROX, 350 nM of each forward primer, 700 nM of reverse primer, 200 nM of probe, 1 X Internal Control Assay (Qiagen, Hilden, Germany), and RNase-Free water. The thermal profile was composed of an initial denaturation step at 95 °C for 5 min, followed by 45 cycles of denaturation at 95 °C for 15 s and annealing/extension at 60 °C for 30 s, as recommended by the manufacturer's instructions of the Master Mix. A fluorescence signal was detected during the annealing/extension step at each cycle.

All of the PCR assays were performed on a StepOnePlus™ Real-Time PCR System, Thermo Fisher Scientific (formerly Applied Biosystems™) (Waltham, MA, USA). At the end of each run all of the samples tested were evaluated for the presence of an amplification curve of the IC, and then evaluated for the presence/absence of a *P.-larvae*-specific target

curve. In the case of the absence of the IC curve, the samples were retested and diluted 1/10 in order to reduce the PCR inhibition factor.

### 2.3.3. Standard Curve and Quantitative TaqMan® Real-Time PCR Assay for the Absolute Quantification of *P. larvae* Spores

A spore suspension of the ERIC I *P. larvae* ATCC 9545 strain in sterile ddH2O was prepared and counted in a Bürker chamber at a light microscope (1000×). Since the approximate spore counting method was based on the use of a Bürker chamber, the count was performed by two independent readers, and the result was estimated as the average value of the two readings (data not shown). The spore suspension ($5 \times 10^9$ mL$^{-1}$) was centrifuged at $9000 \times g$ for 5 min, and the pellet was resuspended with Lysozyme solution, submitted to DNA extraction as described above, and the DNA was stored as stock at $-20$ °C. The number of copies of *P. larvae* chromosomal DNA ($5 \times 10^6$ µL$^{-1}$) in the stock was confirmed spectrophotometrically using the Synergy/HTX (BioTek Instruments, Inc., Winooski, VT, USA) by measuring the absorbance at wavelengths 260/280 nm, assuming the genome size of *P. larvae* to be $4.29 \times 10^6$ bases, according to the NCBI reference sequence NZ_CP019687.1, as reported in [47], corresponding to approximately 5 fg. The *P. larvae* ATCC 9545 strain, with intermediary genome dimension between ERIC I and II as described in Djukic et al., 2014 [48], was used in this study.

All the quantitative TaqMan® real-time PCRs (qPCRs) were performed using a Thermo Scientific Luminaris Color Probe High ROX qPCR Master Mix (Thermo Fisher Scientific, Waltham, MA, USA). The amplification reactions were carried out in a final volume of 25 µL containing 5 µL of DNA extract, 1× Luminaris Color Probe High ROX qPCR Master Mix, 450 nM of each forward primer, 900 nM of reverse primer, 200 nM of probe (Table S1), and RNase-free water. The thermal profile was composed of a uracil–DNA glycosylase (UDG) pretreatment at 50 °C for 2 min and an initial denaturation step at 95 °C for 10 min, followed by 45 cycles of denaturation at 95 °C for 15 s and annealing/extension at 60 °C for 1 min, according to the manufacturer's instructions of the Master Mix. The runs were performed on a StepOnePlus™ Real-Time PCR System. A fluorescence signal was detected during the annealing/extension step at each cycle.

To determine the limit of detection (LOD) of the qPCR assay, five replicates of two independent runs of 10-fold serial dilutions in nuclease-free water of stock DNA, ranging from $3 \times 10^6$ to 0.03 genomes per reaction (in 5 µL), were carried out. The last dilution showing a 100% response was accepted as the LOD and resulted in 0.3 genomes per reaction at a mean Ct value of 34.13.

Samples of DNA that turned out to be *P.-larvae*-positive, based on presence/absence in the RT-PCR assay described in the above paragraph, were tested in three replicates by absolute qPCR, including in each run serial dilutions of *P. larvae* stock DNA, ranging from $3 \times 10^6$ to 3 genomes/reaction, in three replicates as standards. Using StepOnePlus™ Software v2.3, the standard curve showed efficiency and linearity of the reaction with a slope of $-3.365$ and an $R^2$ of 0.999, respectively (Figure 1). The *P. larvae* genome loads in the sample were calculated from standard curve plots: a linear regression analysis between the initial amount of the template and the value of Ct, normalized to the sample weight, and expressed as *P. larvae* genome copies per gram of sugar (Figure 1).

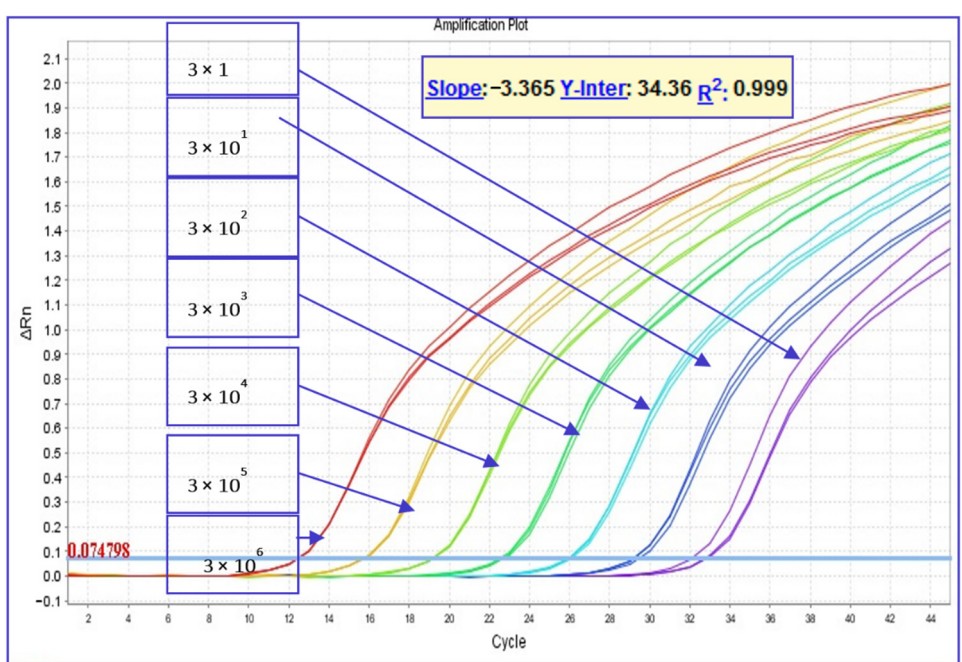

**Figure 1.** Amplification plot of the *P. larvae* genome standard curve. Relation between serial dilutions of DNA from *P. larvae* ATCC 9545 and the threshold cycle (Ct). The Ct values increased in proportion to initial DNA template quantities at a range from 0.3 to $3 \times 10^6$ genome copies. Standard curves showed values of efficiency and linearity of the reaction (slope = $-3.365$; $R^2 = 0.999$).

### 2.3.4. ERIC Genotyping

The ERIC genotypes of the *P. larvae* strains isolated from the samples were defined by ERIC-PCR. For this purpose, the PCR protocol followed was that described in Bassi et al. (2015) [12], using the primers described by Genersch and Otten (2003) [49].

### 2.4. Data Analysis

McNemar's test was used to compare the test results, classified in the presence/absence of *P. larvae* spores. Descriptive statistics were provided regarding the number of colonies and spores, grouped in different contamination levels. Microsoft Excel for Windows and R 4.0.2 (33) were used to manage and analyze the data.

## 3. Results

The presence/absence of *P. larvae* spores by the two tests is reported in Table 1. All of the samples tested by real-time PCR produced a reliable IC value; therefore, they showed no inhibition factors and did not have to be tested diluted.

**Table 1.** Presence/absence of *P. larvae* according to molecular and microbiological analyses of 105 powdered sugar samples.

|  | Culture (+) | Culture (−) |  |
|---|---|---|---|
| RT-PCR (+) | **51** | 18 | 69 |
| RT-PCR (−) | 6 | **30** | 36 |
|  | 57 | 48 | Tot. **105** |

The proportions of positive colonies were 54% (CI 95% 44.3–64.0, 57/105) with the culture method and 66% (CI 95% 55.8–74.7, 69/105) with real-time PCR. A significant difference was found with McNemar's test ($p = 0.02$). Discordance was observed in 24/105 (23%) samples: 18 samples resulted in being negative by the culture method but positive

by real-time PCR, and six samples resulted in being positive by the culture method but negative by real-time PCR. Coincidence was observed in 81 samples (77%): 51 resulted in being positive and 30 in being negative by both methods.

The spore counts obtained by the culture method and qPCR on sugar samples are reported in Figure 2. In 45 samples out of the 51 positives determined by both methods, qPCR found a greater number of spores than the culture method did.

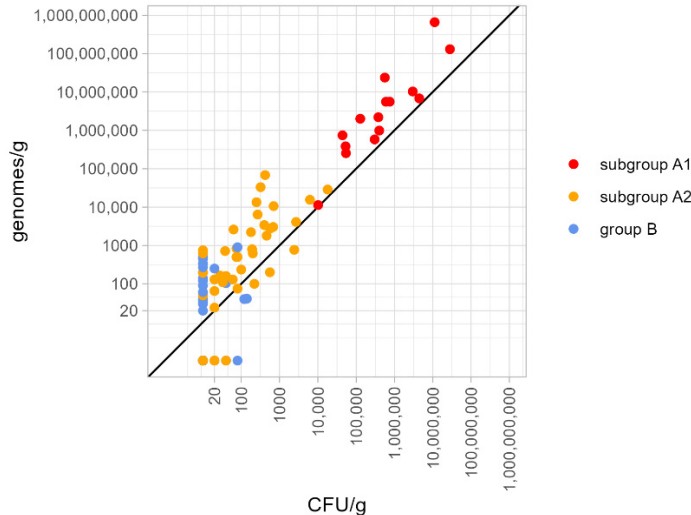

**Figure 2.** Microbiological and qPCR test results of powdered sugar samples grouped by history of AFB's presence in subgroups A1, A2, and B. The diagonal line represents equal counts by the two tests.

In Table 2, spore counts grouped by contamination level are reported. All of the sugar samples of *Sub-group A1* had a spore load greater than $1 \times 10^4$ by the microbiological method, and greater than $1 \times 10^5$ by qPCR. In *Sub-group A2*, the spore load by the microbiological method was less than 100 for 60% (27/45) of the samples, while for 71% (32/45) of the samples by qPCR the spore load was between $1 \times 10^2$ and $1 \times 10^5$. In *Group B*, 82% of the samples had a spore load > LOD by the microbiological method, while 38% of the samples by qPCR had a spore count between 20 and 1000.

**Table 2.** Distribution of results obtained by microbiological and qPCR assays according to the different classes of contamination level expressed in terms of the CFU/g or spore/g of *P. larvae*, respectively.

| Classes of Contamination (CFU/g–Spore/g) | Group A | | | | Group B (*n* = 45) | |
| --- | --- | --- | --- | --- | --- | --- |
| | Sub-Group A1 (*n* = 15) | | Sub-Group A2 (*n* = 45) | | | |
| | Culture Method | qPCR | Culture Method | qPCR | Culture Method | qPCR |
| <20 | 0 | 0 | 11 (24%) | 8 (18%) | 37 (82%) | 28 (62%) |
| 20–100 | 0 | 0 | 16 (36%) | 5 (11%) | 6 (13%) | 8 (18%) |
| 101–1000 | 0 | 0 | 13 (29%) | 18 (40%) | 2 (4%) | 9 (20%) |
| 1001–10,000 | 0 | 0 | 4 (9%) | 8 (18%) | 0 | 0 |
| 10,001–100,000 | 3 (20%) | 0 | 1 (2%) | 6 (13%) | 0 | 0 |
| 100,001–1,000,000 | 8 (53%) | 5 (33%) | 0 | 0 | 0 | 0 |
| >1,000,001 | 4 (27%) | 10 (67%) | 0 | 0 | 0 | 0 |
| **Total** | **15** | **15** | **45** | **45** | **45** | **45** |

Detailed results of all of the samples obtained from both methods are reported in Table S4.

In the apiaries of *Group A* one strain of genotype ERIC I was recovered in only one apiary, whereas in the other five apiary strains of genotype ERIC II were identified. No mixed infections were detected.

In the four apiaries without clinical signs of AFB (*Group B*), the genotype ERIC I and ERIC II were isolated from two apiaries and one apiary, respectively (Table S4).

## 4. Discussion and Conclusions

The low efficiency of germination on culture media of *P. larvae* spores is known [50].

Dingman and Stahly (1983) [51] reported that the highest spore germination on MYPGP agar was about 6% of the direct microscopic spore counts. More recently, Forsgren et al. (2008) [52] showed that different P. larvae genotypes had significant variability in the germination rate on solid media, and Crudele et al. [53] observed that not all the field strains were able to develop in vitro. For these reasons, the culture-based techniques systematically underestimated the number of spores present in the examined samples. The cultivation method detects only a limited percentage of the total spores, those capable to germinate in growing medium. Most of the others are still infectious, therefore potentially capable of causing disease, as not many spores are needed for the onset of the disease in young larvae [2,4,5].

In recent years, many molecular-based protocols have been published for the detection and quantification of *P. larvae* in honey, bees, and debris. They have been used to evaluate the sanitary status of honeybee populations. The PCR-based methods are an alternative to the classic cultivation tests on agar and should also allow for the detection of *P. larvae* spores that do not germinate in vitro, considerably improving the sensitivity.

Conventional PCR has been used to detect *P. larvae* in honey [2,31–33,36,54,55] as well as in beehive debris [35]. Despite numerous advantages, such as higher sensitivity and specificity than the culture method, conventional PCR also has disadvantages and technical limits, because the true quantification of the pathogen is not achievable [45].

Quantitative real-time PCR (qPCR) assay overcomes this limitation of conventional PCR, and it is a time- and cost-effective choice with respect to the cultural assays.

Cultural methods generally underestimate the *P. larvae* spore load because of the weak and inconsistent germination [42,44]. The qPCR protocol allows for a larger quantification of spores, mainly because spores that do not germinate in the culture substrate are also included.

The number of spores found does not necessary translate to infection, as not all spores are viable and qPCR methods would not identify this aspect related to the spore viability.

But qPCR is able to detect sub-clinical levels of *P. larvae* spores. As noted by Forsgren and Laugen [56], the PCR analysis of accumulated winter hive debris was the preferable method for the monitoring of the prevalence of the pathogen irrespective of disease symptoms; while cultural was more accurate for detecting the bacterium in clinically diseased colonies. This may be due to the presence of polymerase inhibitors in some hive materials and/or failures in DNA extraction protocols, among the main factors.

Rossi et al. (2018) [42] proposed to adopt a qPCR protocol to get an early estimation of AFB prevalence, thanks to a reliable quantification of *P. larvae* spores. This could help to prevent the diffusion of *P. larvae* spores from hive to hive, without the need of clinical signs.

Several qPCR methods, designed to detect the 16S rRNA gene of *P. larvae*, have been developed for the detection and/or quantification of *P. larvae* spores in honey [39], in environmental honeybee larva or scales samples [37,38,40], and in honey and hive debris [42]. Unfortunately, all of these methods, although using specific primers targeting the 16S rRNA gene of *P. larvae*, are based on SyBr technology, which requires the analysis of the melting curve of the PCR products at the end of the amplification sessions. This is time consuming and yields vague interpretation.

More recently, new probe-based real-time PCR protocols based on the use of specific fluorescence probes in addition to specific primers have been described.

Dainat et al. (2018) [41] developed a real-time triplex PCR for the qualitative detection of European and American foulbrood in honeybee using primers and a probe designed on the sequence of the tnp60 gene of *P. larvae*.

Kusar et al. (2021) [44] described a novel real-time PCR assay based on the use of a TaqMan® probe detecting the chromosomal metalloproteinase gene, a highly conserved single-copy target, for the presence and reliable assessment of *P. larvae* spores in hive samples. The authors used for the first time digital-PCR technology for absolute quantification of *P. larvae* DNA present in the samples, even in minimal quantities. Digital-PCR technology has the advantage of using the same detection chemistry as qPCR. It allows absolute quantification, without the need for a standard curve. Although digital-PCR is a very promising technology, its use is not yet widespread on a large scale in diagnostic laboratories, like traditional real-time PCR machines.

Here, we describe the use of a new TaqMan® probe-based real-time PCR assay for the detection and quantification of the bacterial spores. To our knowledge, this is the first description of a TaqMan® assay targeting the multiple-copy chromosomal and highly specific 16S rRNA gene of *P. larvae* [2,47]. The choice of the 16S rRNA gene as a target for this new TaqMan® probe-based real-time PCR assay was suggested by the proven robustness of this gene for the detection of *P. larvae*, as previously described [2,29,30]. Furthermore, the presence in multiple-copies (eight copies) of 16S rRNA gene target in the genome of *P. larvae* increases the possibility for the detection of the pathogen in the samples under examination. This aspect is useful and to keep in mind for the development of *P. larvae* diagnostic tests, since a few spores (<10) are enough to cause the arising of the disease.

The new probe-based qPCR method, described in the present work, has been validated for the detection and quantification of the bacterial spores in powdered sugar samples from colonies with different histories of AFB's presence, but it could also be used for the examination of other types of specimens.

The examination of a powdered sugar sample by qPCR can be an effective and useful way for the nondestructive quantitative evaluation of *P. larvae* infection in honeybee colonies, as has been previously described [28]. The results obtained here showed that a significant difference was found between the culture and the qPCR used in favor of the second, which also found a higher level of infection than the culture method did. This is beneficial for the rapid detection of hives infected with *P. larvae*, and it is important for controlling the spread and reducing the impact of AFB. In addition, the results obtained were closely concordant with the clinical–epidemiological conditions of the colonies.

Futures research directions will be focused on to the improvement of qPCR for absolute quantification of *P. larvae* spores. A synthetic *P. larvae* DNA standard could be designed and produced as recombinant plasmid, containing the *P. larvae* target region, according to the available 16S rRNA gene sequences described (Table S2). The synthetic standard will be calibrated using digital-PCR, as previously descried [44], for a reliable and more accurate quantification of *P. larvae* spores in powdered sugar samples, honey, and other hive specimens.

**Supplementary Materials:** The following supporting information can be downloaded at: https://www.mdpi.com/article/10.3390/app12199895/s1, Table S1: Primers and probe of the TaqMan®Real-Time PCR used in the present work; Table S2: Sequences of *P. larvae* 16S rRNA gene from GenBank; Table S3: 16S rRNA gene sequences of Bacterial contaminants from GenBank; Table S4: Details of the results obtained by both methods.

**Author Contributions:** Conceptualization, S.B., E.C. (Elena Carra) and E.C. (Emanuele Carpana); methodology, S.B. and E.C. (Elena Carra); formal analysis, G.G.; investigation, F.B. (Federica Bergamini), G.L., F.B. (Filippo Bosi), S.P. and S.B.; resources, S.B. and E.C. (Elena Carra); supervision, E.C. (Emanuele Carpana); data curation, E.C. (Elena Carra) and S.B.; writing—original draft preparation, E.C. (Elena Carra) and S.B.; writing—review and editing, E.C. (Elena Carra) and G.G.; visualization, E.C. (Elena Carra) and G.G.; funding acquisition, S.B. and E.C. (Emanuele Carpana). All authors have read and agreed to the published version of the manuscript.

**Funding:** This research was partially funded by the Italian Ministry of Health, grant number G88F13000630001 (IZSLER_PRC2013101).

**Institutional Review Board Statement:** Not applicable.

**Informed Consent Statement:** Not applicable.

**Data Availability Statement:** Not applicable.

**Conflicts of Interest:** The authors declare no conflict of interest.

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
