# Peer review of "A Probe-Based qPCR Method, Targeting 16S rRNA Gene, for the Quantification of Paenibacillus larvae Spores in Powdered Sugar Samples"

_applsci, doi:10.3390/app12199895_

Round 1

Reviewer 1 Report

This work is a technical study in which authors compared two different methods for AFB detection and quantification. As we know and is cited in the text, detection of AFB by molecular means has been previously tackled in many studies, whether by conventional PCR or qPCR. Therefore, this study does not provide much of a breakthrough or scientific achievements. The potentially novel aspect of this work is testing a probe-based quantitative real-time PCR for AFB quantification. Unfortunately, the title does not reflect that, nor the main message presented in this manuscript. This should be addressed in a serious manner. From an epidemiological stand point, the number of spores counted or found does not necessary translate to infection as not all spores are viable. This point is crucial for beekeepers and should be incorporated in the text. The qPCR method would not identify this aspect related to the spore viability. The manuscript needs intensive revision for its sentence structure and English language for better readability. Despite the challenge for beekeepers to refer to molecular analyses for disease identification (cost, time, availability …etc), the qPCR technique presented in this study is valid and I do not see major flaws preventing the publication of this work. Due to lack of time, I only highlighted some minor edits below…many more need to be addressed throughout the text.        

L32: a significant difference between the two methods 

L33: rephrase this… out of the 51 positive samples, 45 showed higher infection my qPCR compared to the culture.

L35: change “situation” to “status” 

L142: correct “described”

L144: what was the concentration of the proteinase K used, right it down please. 

L147: what was the final elution buffer? It should be specified. 

L170: was this cycling the recommended one from the manufacturer of QuantiFast Master Mix? 

L192: “corresponding to about to 5 fg”? change to “corresponding approximately to 5 fg”. 

L192-194: Not clear…rephrase this sentence please… something like “a strain with intermediary genome dimension between ERIC I & II was used in this study as described in Djukic et al…”

L311: change or rephrase such as “…sessions, which is time consuming and yield vague interpretation”. 

Author Response

Please, see the Report Notes file sent

Reviewer 2 Report

Dear authors,

I have read carefully your paper and have some suggestions to improve its quality.

Please, write in the Abstract section few sentences about disease caused from P. larvae.

Line 27. Replace different histories of AFB’s presence with different infection levels.

Line 30. Replace 16S rRNA gene with 16S rDNA gene, which codded SSU rRNA.

Please, replace q-PCR with qPCR throughout the text.

Line 38-40. Avoid repetitive words in title of the paper and keywords.

Line 47. diseased dead brood – dead brood.

Line 57. Before this sentence, please, write a short paragraph according the most frequently used diagnostic methods and techniques, and after that continue with the molecular methods.

Line 62. What is the geographic distribution of these genotypes?

There is no any information about molecular methods concerning 16S rDNA gene or gene fragment, please add some data.

Line 98. traditional PCR – conventional PCR.

Line 101. Before the aim of the study the authors may write something about advantages of the method used over other applying methods.

Line 110. How many hives included these apiaries - A and B?

Line 121. Sugar – may be collected sugar is more appropriate.

Line 123. MYPGP agar – what does it mean? Give a full name and after that abbreviation in brackets.

Line 130. TSYEA???

Line 146-147. Any evaluation of purified DNA – gel electrophoresis, spectrophotometrically?

Line 153. I cannot see any ERIC genotype in Table S1.

Line 164. Please, replace RT-PCR with qPCR due to the reason that RT-PCR may denotes reverse transcription polymerase chain reaction.

Line 170. Replace activation step at 95°C with initial denaturation.

Line 181. ATCC 9545 strain or genotype? In their paper authors mentioned only genotype and not strain.

Figure 1. chromosomal DNA from P. larvae ATCC 9545 – this is completely wrong, why authors used chromosomal DNA?

Line 250. Replace concordance with coincidence.

Line 256. 10.000 – I would prefer 1x104

Line 275. Recovered – Identified.

Line 274-278. Whether the authors sent any obtained sequence to GenBANK database?

Line 285. in vitro – change with in vitro.

Line 295-297. Please, see the following papers:

Salkova D, Shumkova R, Balkanska R, Palova N, Neov B, Radoslavov G, Hristov P. Molecular Detection of Nosema spp. in Honey in Bulgaria. Veterinary Sciences. 2022; 9(1):10. https://doi.org/10.3390/vetsci9010010

Ribani, A., Utzeri, V. J., Taurisano, V., & Fontanesi, L. (2020). Honey as a source of environmental DNA for the detection and monitoring of honey bee pathogens and parasites. Veterinary sciences, 7(3), 113.

Ribani, A., Utzeri, V. J., Taurisano, V., Galuppi, R., & Fontanesi, L. (2021). Analysis of honey environmental DNA indicates that the honey bee (Apis mellifera L.) trypanosome parasite Lotmaria passim is widespread in the apiaries of the North of Italy. Journal of Invertebrate Pathology, 184, 107628.

The Discussion section needs more work. Authors should discuss their results against previous similar studies, looking for an explanation to support or reject the results.

Also, the reference list does not follow requirements of the Journal.

Author Response

Please, see the report notes file sent

Round 2

Reviewer 2 Report

Dear Editor,

The authors significantly improve their submission from the first round and I suggest to accept this paper.

Author Response

The authors would like to appreciate reviewer# 2's positive comment.